# Self-Assembling Behavior of pH-Responsive Peptide A_6_K without End-Capping

**DOI:** 10.3390/molecules25092017

**Published:** 2020-04-26

**Authors:** Peng Zhang, Fenghuan Wang, Yuxuan Wang, Shuangyang Li, Sai Wen

**Affiliations:** School of Light Industry, Beijing Technology and Business University (BTBU), Beijing 100048, China

**Keywords:** self-assembly, pH-responsive peptide, molecular dynamics simulation, nanoribbon, β-turn

## Abstract

A short self-assembly peptide A_6_K (H_2_N−AAAAAAK−OH) with unmodified N− and C−terminus was designed, and the charge distribution model of this short peptide at different pH was established by computer simulation. The pH of the solution was adjusted according to the model and the corresponding self-assembled structure was observed using a transmission electron microscope (TEM). As the pH changes, the peptide will assemble into blocks or nanoribbons, which indicates that the A_6_K peptide is a pH-responsive peptide. Circular dichroism (CD) and molecular dynamics (MD) simulation showed that the block structure was formed by random coils, while the increase in β-turn content contributes to the formation of intact nanoribbons. A reasonable explanation of the self-assembling structure was made according to the electrostatic distribution model and the effect of electrostatic interaction on self-assembly was investigated. This study laid the foundation for further design of nanomaterials based on pH-responsive peptides.

## 1. Introduction

In recent years, with the continuous development of solid-phase synthesis technology, self-assembled synthetic peptides have attracted increasing attention in many fields [1,2,3]. Due to its flexible design, easy synthesis, good biocompatibility, and easy biodegradation, it has great application potential as an antibacterial agent [4,5], a carrier for controlled-release drugs [6,7], and cell culture materials [8,9].

Among the self-assembling peptides, heptapeptide A_6_K (Ac−AAAAAAK−NH_2_) has been extensively studied for self-assembly features and shows promising future in clinical use. The surface-active peptide A_6_K was first reported to undergo self-assembly in water by G. Von Maltzahn et al. [10]. Transmission electron microscopy and dynamic light scattering experiments have proved that it can form nanotube structures when the pH is below the theoretical pI (isoelectric point) value. Consequently, they proposed potential applications of A_6_K in surfactants, biotechnology, and nanotechnology. Nagai et al. followed to report the critical aggregation concentration (CAC) of A_6_K in water and PBS, and observed the nanotube structure under an atomic force microscope (AFM) [11]. To further understand the dynamics of self-assembly, Wang et al. used a combination of TEM, AFM, and small angle neutron scattering (SANS) to demonstrate the self-assembly structure of A_6_K in water at different times. In the first 1 h of self-assembly, the peptides stacked into blocks and gradually assembled into columnar fibers within the next 24 h. In the following two weeks, the assembly structure remained as nanofibers without significant changes. This study provided insight into the change of peptide self-assembly action in time latitude [12]. Since A_6_K can self-assemble to form vesicles in water, related reports have shown that it has a good application prospect in the packaging and delivery of hydrophobic drugs [13,14]. As an antimicrobial peptide (AMP), it shows antimicrobial activity against Gram-negative *Escherichia coli* and Gram-positive *Staphylococcus aureus* [15]. In addition, A_6_K can promote the selective adsorption of specific substrates and enhance photovoltaic performance in the application of biological photovoltaic cells. These studies have great significance for the development of new solar cells and photocatalysts [16,17].

Some researches concerning the effect of end-capping of A_6_K (Ac−AAAAAAK−NH_2_) on self-assembled structures had been carried out recently. Bucak et al. modified the A_6_K with trifluoroacetic acid (tfa) to synthesize A_6_K (tfa)_2_ peptide. This peptide can self-assemble into nanotubes in water, but has no surface activity [18,19]. Qiu et al. studied the variant A_6_K^±^ (Ac−AAAAAAK−OH). They removed the protective group at the C−terminus of A_6_K and exposed the carboxylic acid group. The A_6_K^±^ has a lower critical micelle concentration than the capped one, and the self-assembly situation is also different. A_6_K^±^ forms short nanofibers at low pH, longer fibers at pH 5, and nanospheres at high pH [20]. Their results showed that the variation of the groups at the two terminal ends of peptide may significantly affect its self-assembly structure. However, the mode of self-assembly of A_6_K peptide without end-capping remains veiled. Therefore, uncapped A_6_K peptide (H_2_N−AAAAAAK−OH) was designed in this study. We found that the self-assembly of this A_6_K peptide is pH-responsive. It is inferred that amino acids at both ends have different pKa values, which will cause the peptides to have different charge distribution states at different pH. In this study, we built a charge model through simulation calculations. Based on the model, molecular dynamics simulations and experiments were implemented to investigate the self-assembly structure, and the application prospects of uncapped A_6_K were discussed.

## 2. Results and Discussion

### 2.1. Electrostatic Potential Energy Distribution

The pKa values of the C-terminal, N-terminal, and lysine residue side chains of A_6_K were obtained on the APBS-PDB2PQR server (Figure 1B). According to the pKa value, the electrostatic potential energy was simulated at pH 2, 7, 8, and 11, respectively. Based on the results, we established four models with different electrostatic potential energy distributions (Figure 1C). In these models, the pH increased from left to right, and the total net charge of A_6_K was 2e, 1e, 0, and -1e, respectively. At pH 2, the amino group of peptide A_6_K was protonated, and the carboxyl group was substantially uncharged. As pH value of 7 was greater than the pKa value of the C-terminal carboxyl group, consequently, the carboxyl group was negatively charged, while the N-terminal amino group was still positively charged. When the pH value was 8, which was between the pKa values of the N−terminus of the peptide and the lysine residue, the N−terminus of peptide was almost uncharged and the net charge of A_6_K was zero. In fact, after two weeks of incubation in the solution, white precipitates were observed in both pH 8 and 9 samples, implying that the peptide had an isoelectric point between pH 8 and 9. The A_6_K isoelectric point calculated by the online analysis tool ExPAsy was 8.8, which is also in line with our prediction results [21]. Subsequently, the self-assembly morphology at different pH and the secondary structure of A_6_K assembly were studied in combination with the charge model described above.

### 2.2. Self-Assembling Structure

In the preliminary research on A_6_K (H_2_N−AAAAAAK−OH), we found that self-assembly could occur in water at 10 mM. At neutral pH there was a tendency to form a band-like structure, but it was not complete. So, we increased the sample concentration to 16 mM in this research. To investigate the effect of pH on self-assembly of A_6_K more comprehensively, we prepared a series of aqueous solutions of peptides from pH 2 to 11, instead of being limited to the four pH values in the charge distribution model. Figure 2 presents the TEM image of the A_6_K peptide solution at different pH. Blocks with clear boundaries but irregular shapes were observed in samples with a pH value of 2, while irregular blocks observed with pH values of 3 and 4 had no clear boundaries as in pH 5. The images at pH 5 and 6 showed that although the band-like structure began to appear, the number was extremely scarce, and most of the areas under the electron microscope were loose peptide stacks. At pH 7, a large number of stacks (red circle) extended to form a band after fusion. At pH 8, the formation of supramolecular assembly structures could be seen, and nanoribbons up to a few microns in length could be easily found. It is also worth noting in the figure that the nanoribbons could merge with each other, widen and bend, as shown by the red circle. Broken nanoribbons and block structures are shown at pH 9 and 10. Compared with assembly structures at pH 10, the band formed at pH 9 was longer and the structure was more complete. The last electron microscope image showed that the nanoribbons disappeared at pH 11, and most of the peptides self-assembled to a block structure. It has clearly shown that the self-assembly process of A_6_K was affected by pH, which proved that our designed A_6_K (H_2_N−AAAAAAK−OH) was a pH-responsive peptide. With the change of pH from low to high, the self-assembled morphology altered from a block stack to a nanoribbon, and finally switched back to a block stack.

The self-assembly results observed with electron microscopy can be interpreted with the charge model. According to the charge distribution models of pH 2 and 11 (Figure 1C), the peptide A_6_K only had a positive charge of 2e and a negative charge of 1e, respectively. When peptides molecules were close to each other, due to the influence of electrostatic repulsion, it was difficult for the peptides to aggregate into a large, tightly ordered structure. Therefore, there was no structural basis for the formation of nanoribbons. Indeed, the formation of nanoribbons was not observed under a transmission electron microscope. In contrast, we observed the nanoribbons at pH 7 and 8. According to the charge model, A_6_K had both positive and negative charges at these two pHs. The net charge at pH 7 and 8 were 1e and 0, respectively. Therefore, electrostatic attraction and repulsion between peptide molecules coexisted. Through the change of the molecular space position, peptides closely aligned by electrostatic attraction and maintained a certain structure under the balance of attractive-repulsive force. The coexistence of two kinds of charges provides the possibility for A_6_K self-assembly to form nanoribbons. Comparing the structures formed at pH 7 and 8, the nanoribbons at pH 8 were longer, wider and the interconnected supramolecular structure was also found. This may be due to the electrostatic repulsion at the N−terminus of the peptide. On the other hand, the alanine tails should draw together under hydrophobic interaction, but the positive charges at the N−terminus of the peptides repulse each other at pH 7, making aggregation difficult.

### 2.3. Secondary Structure

Circular dichroism is an effective method for rapid determination of the secondary structure of proteins and peptides [22,23]. Figure 3 shows the circular dichroism (CD) spectrum of A_6_K at different pH. There were obvious negative peaks around 198 nm at pH 2–7, which is a characteristic of random coil formation. Notably, the CD spectrum changed significantly at pH 8. Besides a negative peak near 198 nm, a clear positive peak also existed at 205 nm, which indicated that there was equilibrium between random coil and β-turn conformation. On the other hand, the predominance of a weak positive peak around 200 nm at pH 9, and a negative peak at 198 nm at pH 10 and 11 also shows the presence of a random coil. Therefore, we inferred that the secondary structure was changing gradually from β-turn into random coil at pH 9.

Combined with the TEM results, it was demonstrated that the self-assembled structure of the block stack was based on random coil, and the formation of complete nanoribbons required participation of a certain amount of β-turn. Since nanoribbons were formed under the influence of hydrogen bonding, electrostatic, hydrophobic interaction, and specific secondary structures, the band structures that appeared at pH 5–7 were not complete. To visually verify whether the β-turn was related to the formation of nanoribbons, molecular dynamics simulations were further applied.

### 2.4. Molecular Dynamics Analysis

The method of molecular simulation can unveil the early situation of self-assembly. The simulation results of A_6_K (Ac–AAAAAAK–NH_2_) by Sun et al. showed that the peptides mainly used loosely packed disordered curly aggregates, which could form monolayer lamellas [24]. In our simulation results, as shown in Figure 4A,D, A_6_K (H_2_N–AAAAAAK–OH) formed a loose agglomerate structure and was distributed in water. Although the aggregates shown in Figure 4B also existed as agglomerates, they were denser and larger. A helical extending aggregate was observed in Figure 4C. The simulation results also show that the self-assembly process was completed under various interactions. The Figure 4E further illustrates the helical columnar structure formed by the aggregates in a defined extension direction axis at pH 8. Alanine (red part) as the hydrophobic part of the peptide was close to each other to form the backbone of the aggregate. The hydrophilic lysine (green part) was distributed outside the backbone, which more visually shows the hydrophobic effect in self-assembly. As we have inferred previously, because the peptide only had the same kind of charge at pH 2 and 11, it was difficult to form large aggregates under the influence of electrostatic repulsion. In contrast, there were both positive and negative charges at pH 7 and 8. It is easier to form large aggregates under the influence of electrostatic attraction. Under the action of electrostatic attraction, molecules are more likely to get closer to each other. At the same time, the intermolecular hydrogen bonding can make them stably gather together, which provides an important basis for the assembly of nanoribbons into larger structures. Since the N−terminus of the peptide was not charged at pH 8, the electrostatic repulsion force was smaller than that of the N−terminus at pH 7, so the aggregate formed was larger.

The secondary structure plays an important role in the self-assembly process, and it is difficult for peptides to form compact structures without turns. The β-turn was discovered by using CD to detect the sample, which self-assembled to form nanoribbons. We analyzed this result in combination with molecular simulation. It should be noted that β-turn is only one type of turn, so they were counted separately. In the STRIDE algorithm, only turn is used for statistics, so β-turn needs to be further counted [25]. There are two common classifications of β-turn. One is to form a hydrogen bond between the C=O of i residues and the N-H of (i + 3) residues in the main chain, and the other is that the Cα distance between residues i and (i + 3) is less than 7Å [26,27]. Visual molecular dynamics (VMD) was used to filter each turn structure in the simulation results according to the classification. The turn and β-turn count results and their percentages are shown in Figure 4. The counting results showed that the contents of turn and β-turn reached 46% and 16% at pH 8, respectively. This value was higher than other conditions, and was consistent with the results of circular dichroism. It is shown that the increase of β-turn structure was conducive to the formation of helical columnar structures. We inferred that based on the columnar helical structure, nanoribbons could be further formed.

Based on the results of our research, several application hypotheses were proposed. Unfortunately, after measuring the uncapped A_6_K sample, unlike the capped ones, we found that it had almost no surface activity. We also performed fluorescence experiments with the hydrophobic dye pyrene, and have not found that the characteristic absorption peak of pyrene was significantly enhanced, which proved that pyrene was not encapsulated in the self-assembly of the peptide. In summary, we believed that this unmodified A_6_K is not suitable as a delivery vehicle for hydrophobic drugs or as a biosurfactant. Fortunately, this non-surface-active amphiphilic peptide is pH-responsive and the exposed amino and carboxyl groups provide convenient binding sites. Inspired by Jungok Kim et al., we believe that this unmodified A_6_K may be suitable for the development of new gold nanomaterials [29]. They combined peptides with gold to produce gold nanostructures, and the structure can be regulated by changing pH. This ability makes it potentially useful in the fields of nanoelectronics, sensors, and optoelectronics. Here we only studied the self-assembly process of peptides in aqueous solution. However, the self-assembly of peptides in buffers with a large number of ions may be different. According to Valeria Castelletto’s research, peptides self-assemble into β-sheet nanotape in water, but in phosphate-buffered saline (PBS), the nanoribbons associate to form a hydrogel due to charge screening [30]. Therefore, the self-assembly of A_6_K in PBS and whether it can form a hydrogel could be investigated in future. In addition, many studies have shown that the secondary structure that forms nanoribbons is a β-sheet [31,32,33]. Quite the opposite, our experimental results showed that the nanoribbons formed by A_6_K were based on β-turn, which expand the understanding of the construct of nanoribbon structures.

## 3. Materials and Methods 

### 3.1. Samples

The desalted of peptide A_6_K (H_2_N–AAAAAAK–OH) with both termini unblocked was synthesized commercially by GL Biochem (Shanghai, China) Ltd. and the purity is determined by the supplier >95%. A_6_K has a molecular weight of 573 g mol^−1^. Figure 1A shows the structural formula of the designed peptide A_6_K. For the sample preparation, in order to ensure that the peptide concentration is sufficient for self-assembly, a 16 mM solution was prepared. According to the calculation of the molecular mass of the peptide, the powder of the peptide and Milli-Q water (18 MΩ cm) were quantitatively weighed. After vortexing for ca. 30 s and sonicating for 15 min with a bath, the peptide was completely dissolved and a clear aqueous solution was obtained. The clear aqueous solution was separated into a centrifuge tube and adjusted the pH in each tube with NaOH or HCl accordingly. The solution was incubated at room temperature for at least 2 weeks to ensure the complete assembly before characterization.

### 3.2. Charge Model

AmberTools15 was used to make a conformational model of the short peptide A_6_K [34]. The online analysis tool APBS-PDB2PQR was used to calculate the electrostatic potential of short peptides with default settings at different pH [35,36]. The calculated results were observed with PyMol, version 1.8.

### 3.3. Transmission Electron Microscopy (TEM)

TEM images were obtained on a FEI Tecnai Spirit D1266 transmission electron microscope (FEI, Eindhoven, Netherlands). Carbon-coated copper grid was hydrophilized by glow discharge to improve wetting properties. Of the sample solution 5 μL was placed on a copper mesh, and after adsorption for about 1 min, excess liquid was removed with filter paper. A drop of a 2 wt% aqueous solution of uranyl acetate was used to stain the grid, and it was evaporated to dryness in air before observation of the sample with a transmission electron microscope.

### 3.4. Circular Dichroism Spectroscopy (CD)

CD measurements were performed using a MOS-500 spectrometer (BioLogic Science Instruments, Claix, France) at room temperature. A 0.1 mm quartz cell was used to measure the samples after two weeks of incubation. The scanning speed was 60 nm/min and the recorded scanning range was 190–250 nm. Response time was 1 s and data pitch was 1 nm. Spectral smoothing was performed to reduce the noise by using Bio-Kine 4.74 software.

### 3.5. Molecular Dynamics (MD) Simulation

Using GROMACS 5.0.4 software package, OPLS-AA/L full atomic force field and tip4p water molecular model were selected for explicit water model molecular dynamics simulation [37,38,39,40]. Eighty peptides were randomly placed in a cubic box with a volume of 512 nm^3^, and filled with water molecules and counter ions. The boundary of the box was processed periodically. The steepest descent method was used to relax the initial structure by 50 ps before the simulation. The NVT ensemble is performed at a temperature of 300 K for 100 ps on the system, and then performed an NPT ensemble for 20 ns. For calculation, the bond length of the chemical bond was fixed using the Linear Constraint Solver (LINCS) algorithm. Long-range electrostatic interactions were calculated using the Particle Mesh Ewald (PME) algorithm, and short-range electrostatic interactions were calculated using the Coulomb formula. Both the truncation radius and the van der Waals effect were 1.4 nm, and the calculation time step was 2 fs. The Nose–Hoover thermostat was implemented to maintain the system at 300 K. The system pressure was controlled by the Parrinello–Rahman method with a constant pressure of 1 bar, and the pressure constant was 1 ps.

## 4. Conclusions

In this study, the self-assembly process and influencing factors of pH-responsive peptide A_6_K (H_2_N–AAAAAAK–OH) was investigated. By changing the pH of the solution, the A_6_K peptides present different charge distributions. TEM images proved that under different charge distributions, the peptides assembled into different structures. Intact nanoribbons were found near the isoelectric point, and changed into blocks at lower or higher pH. CD and simulation results indicate that β-turns promote the formation of nanoribbons. The results of simulation also indicate that the self-assembly structure of the peptide is affected by hydrogen bonding, electrostatic, and a hydrophobic interaction. This study provides an opportunity for the development of novel nanomaterials based on pH-responsive peptides.

## Figures and Tables

**Figure 1 molecules-25-02017-f001:**
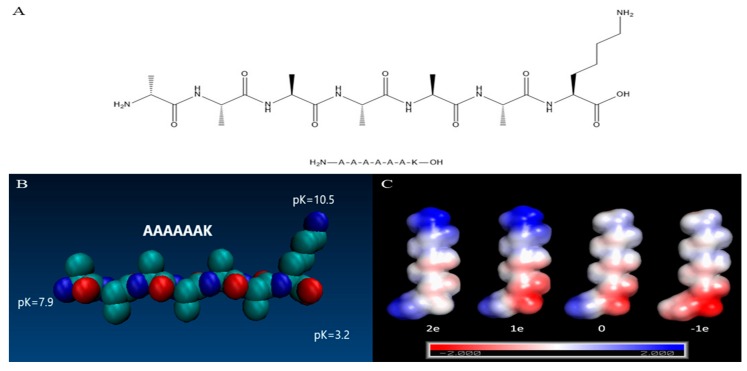
(**A**) Structure of A_6_K. (**B**) Space filling model of A_6_K peptides. pKa values for charged groups were calculated using APBS (adaptive Poisson–Boltzmann solver). (**C**) Electrostatic potential (kBTe (eV)) mapped onto the solvent accessible surface of the A_6_K peptide at pH = 2, 7, 8, and 11 (left-to-right, correspondingly). The net charge of A_6_K is written under the model.

**Figure 2 molecules-25-02017-f002:**
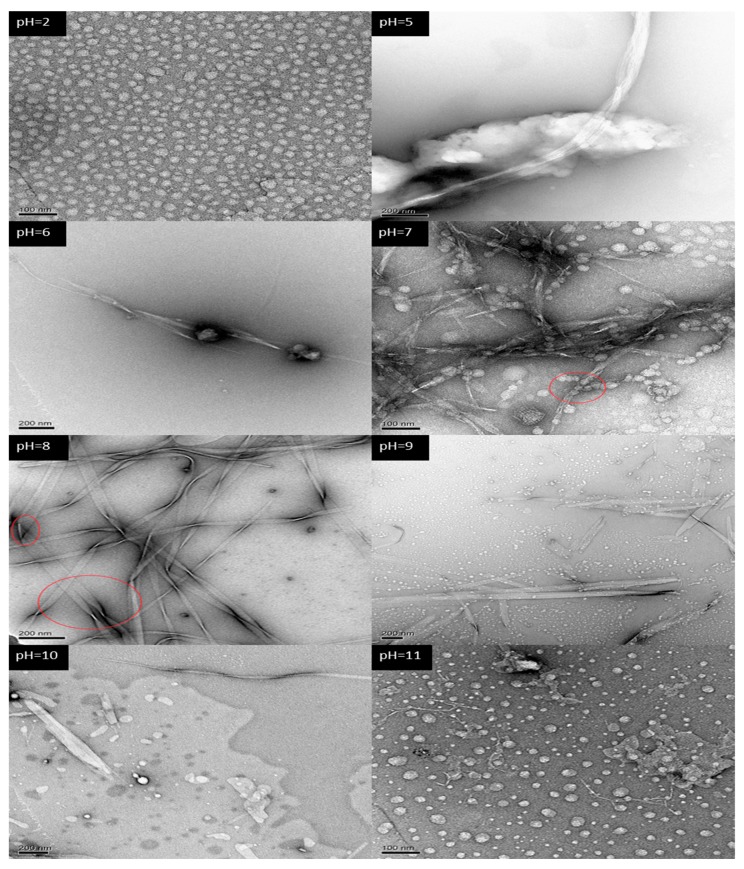
TEM morphologies of the self-assembled structures of A_6_K at different pH. The label in the upper left corner corresponds to different pH values.

**Figure 3 molecules-25-02017-f003:**
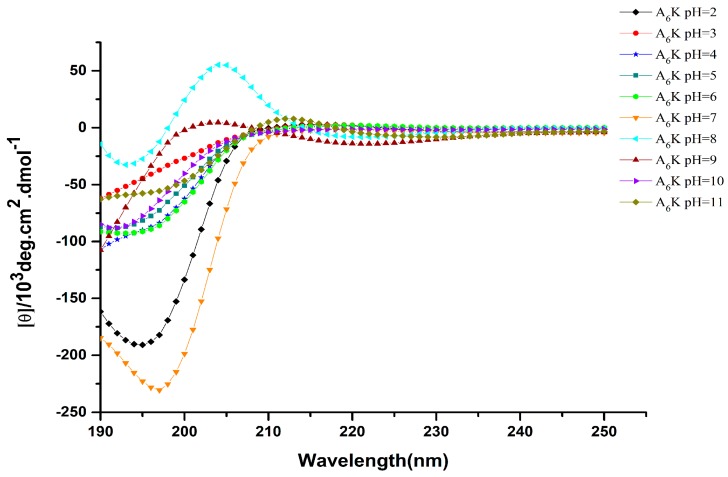
Circular dichroism (CD) spectra of 16 mM peptide A_6_K at different pH.

**Figure 4 molecules-25-02017-f004:**
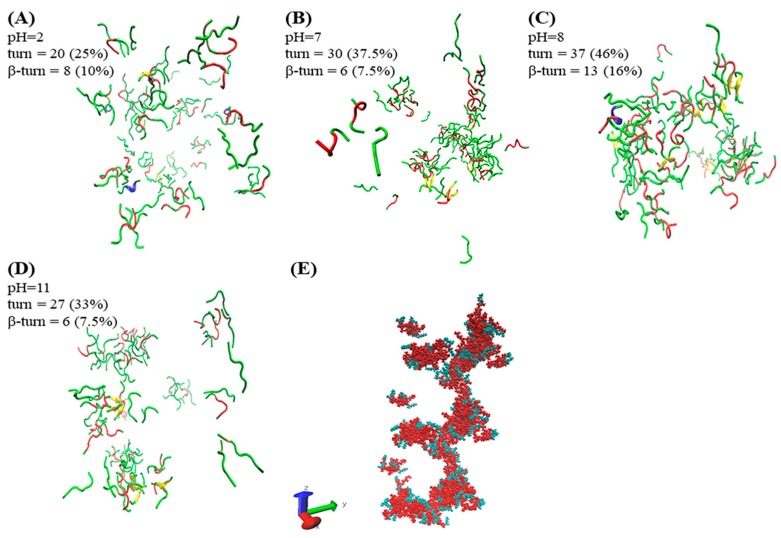
Final snapshots (at time = 20 ns) of assemblies for A_6_K were generated with visual molecular dynamics (VMD) at different pH [28]. The water and ions are not shown for better clarity of the figure. (**A**–**D**) shows the simulation results using the secondary structure in VMD (the secondary structure algorithm uses STRIDE). The secondary structures of turn, helix, coil, and sheet are represented by red, blue, green, and yellow, respectively. (**E**) shows the periodic display of the aggregates in the z-axis at pH 8.

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
