# Peer review of "Self-Assembling Behavior of pH-Responsive Peptide A6K without End-Capping"

_molecules, 2020, doi:10.3390/molecules25092017_

Round 1

Reviewer 1 Report

In this work the authors conduct a study on the self assembled structures of the amphiphilic peptide A6K in aqueous solution at different pH ranging from 2 to 11 by TEM microscopy, CD and molecular dynamics analysis. They show that, as the pH changes, the A6K peptide can assemble into blocks or nano ribbons, which indicates that the peptide is pH-responsive. The secondary structure of the peptide is assigned by CD spectra that indicate that random coils prevail in the block aggregates while the increase in beta-turn content contributes to the formation of the nano ribbons. Furthermore, the effect of electrostatic interactions on self-assembly is investigated.

The proposed study provides molecular insights into the self-assembly process of the A6K peptide at a microscopic level, which might be helpful for the design of novel nanostructure formed by amphiphilic pH-responsive peptides for targeted applications. Moreover, the self-assembly process of A6K is similar to the processes of formation of many biological filaments such as collagen fibrils, lipid tubules and amyloid fibrils [Soft Matter, 2009, 5, 3870-3878]. For this reason the topic is of relevant interest also for the comprehension of many processes of aggregation of peptides involved in neurodegenerative diseases such as Abeta-42. 

In my opinion the work can be published in ‘Molecules’ journal after major revisions that I suggest as follows:

- The scale bar of the TEM images in Figure 2 should be enlarged and uniformed in unit of measure for better clarity.

- In Figure 4 the A6K secondary structure results at pH 5,6,9,10 are missing and they should be added for completeness of data.

- The assignment of secondary structure of the peptide should be corroborated by other techniques such as IR or Raman spectroscopy for a precise attribution of the beta structure present in the nano ribbons. Several authors previously performed conformational analysis of A6K oligomers at pH 6 indicating that A6K peptide chains predominantly adopt random coil structure ad to a much lesser extent of beta-sheet conformation, represented by beta-strands in antiparallel alignment stabilised by hydrogen-bonding interaction. [Biomacromolecules 2015, 16, 9, 2940-2949]

- An extended explanation of the hydrogen bonding interaction involved in the calculation of A6K secondary structure might clarify how beta-strands can be stabilised to form nano ribbons structures.

Reviewer 2 Report

The manuscript of Zhang et al. describes the studies performed on the self-assembly and structure of the short peptide H2N-AAAAAAK-OH at different pH values. In the introduction, the state of the art of this short peptide is properly described. All the previous studies on the self-assembly of this peptide have used protections at the N-terminus and/or C-terminus of the peptide. The authors have designed the uncapped A6K peptide (H2N-AAAAAAK-OH) to study its self-assembly. Both experimental (transmission electron microscopy and circular dichroism) and computational methodologies have been used.

The results are well presented and the different structures observed are well rationalized in base to the molecular changes at different pH values.

I would just like to ask the authors if there have been any issue with the solubilization of the peptide (16 mM is pretty concentrated for a peptide) and if sonication was enough for disrupting any aggregated structures before preparing the solutions.

Do you have any data to correlate the kinetics of self-assembly? In the experimental section (line 207) it is stated that the samples were incubated for 2 weeks to ensure complete assembly. Is this the case for all the pH values? Providing data on the kinetics of the self-assembly would be very interesting as well.

Reviewer 3 Report

Reviewer #1: Zhang et al. has described the self-assembly pH dependence and structural features as well for a new synthetic unmodified A6K peptide. A set of experiment were carried out in different pHs and the author prove that self-assembly process in favorable in pH 7 and 8, near to the peptide isoelectric point. The results adequately support the pH dependence on the self-assembly process. However, the structure characterization and discussion need to be supported by other data, as suggested in the major revisions.  Although the manuscript is well organized and written, a precise revision in the writing is required as pointed out in the minor revisions. Considering the relevance of describing novel self-assembly peptide, this reviewer recommends the publication of the work in the Molecules after consideration of the following major and minor points:

Major points:

In the item 2.1 ln78 the authors conclude that observed peptide precipitation indicate isoelectric point close to 8? The author need to comment on the text which solution condition (pH) they are referring. It was observed precipitation in lower and high pH?

Figure 3. The spectra coloring of Figure 3 not clearly differ the pH condition. In that case, the author need to change color or even use symbols in order to provide a better distinction pattern. Please, include the peptide concentration in the figure title.

Indeed, the key issue of this work is related to the interaction forces, which provide a defined self-assembly in pH 7 and 8. In the Molecular dynamics analysis (item 2.4.), first paragraph, the authors argue about self-assembly structure at atomic level take in account the Figure 4C. As expected, a supramolecular arrangement with an hydrophobic core composed by Ala residues and hydrophilic surface composed by exposed charged Lys side-chains was observed. However, some empirical evidence is required in order to reinforce this structural proposition, since CD structure support only b-turn conformation, For instance, Zeta potention measurenments in each pH probably should reveals higher values at pH 7 and 8.

Item 3.1 – The author must describe in details de HPLC (column, flow, colume injection, sample concentration, wavelength, solvents…) and ESI (ionization potential, negative or positive mode…) experimental condition.

Item 3.1 – The author must describe how the peptide concentration (16 mM) was determined.

Item 3.4 – Include the CD experiment data: scan speed, data pitch, response time and accumulations. Please, also include the software used for CD spectra analyses.

The author must give some explanation about chosen peptide concentration in the beginning of Results and Discussion. Did the author developed a previous study of critical concentration for self-assembly process?

In the conclusion, the authors seem correlate CD data with  hydrophobica and electrostatic effects, please rewrite this sentence “The results of circular dichroism chromatography and molecular simulations revealed that self-assembled structures were mainly affected by hydrophobic, electrostatic effects, and secondary structure of the peptide.”

Why the authors do not estimate the b-turn content from the CD spectra?  

Minor points:

P1ln12 – remove the coma of sentence - “model, and the corresponding …”

P1ln18 – remove the coma of sentence – “model, and the effect of electrostatic i”

P2ln68 – There is no label B or C in the figure 1 – The respective labels must be inserted in this figure.

P2ln74 – change the sentence – “…group, the carboxyl group…” to “…group, consequently, the carboxyl group…”

P2ln77 – change the sentence – “In fact, white precipitation did occur in the peptide…” to “In fact, white precipitation is observed in the peptide…”

P3ln89 – change the word “TEM photograph” to “TEM image”

P3ln96 – The author must to include a reference for the sentence “Significantly, it is common that nanoribbons can fuse with each other, become wider and bent as shown in the red circle.”

P4ln107 – change the sentence “When peptides are…” to “When peptides molecules are…”

P3ln91 – The authors are invited to present TEM images for both pH 3 and 4 in supplementary information, since they are described in the text. Alternatively, the author could mention similarity with pH 2 or pH 5, which are presented in the Figure 2.

P4ln114 – Naturally, in all pH studied, electrostatic attraction and repulsion coexist. Probably, near to isoelectric point there is equilibrium between the two forces. In that way, the author are invited to rewrite the sentence “In this state, electrostatic attraction and repulsion coexist.”

P4ln118 – What does “more complete strucutre” means? Could it be changed to extended or interconnected supramolecular structure, for example?

P4ln130 – change the sentence “which indicated that random coil and β-turn angle coexist. There is a weak positive peak around 200 nm at pH 9, and a negative peak at 198 nm also shows the presence of random coil at pH 10 and 11.” to “which indicated that there is an equilibrium between random coil and β-turn conformation. On the other hand, the predominance of weak positive peak around 200 nm at pH 9, and a negative peak at 198 nm at pH 10 and 11 also shows the presence of random coil.”

P5ln147 – Change the sentence “is mainly carried out under the action of hydrophobic and electrostatic.” to “is dependent of both hydrophobic effect and electrostatic attraction.”

P5ln148 – Change the sentence “aggregates in the extension direction” to “aggregates in a defined extension direction axis”

P8ln245 – Change the sentence “results of circular dichroism chromatography…” to “results of circular dichroism spectroscopy…”

Round 2

Reviewer 1 Report

Dear authors,

thank you for providing a revised form of the paper taking into account of my comments and suggestions.

Although some issues remain still open and not completely investigated, such as the precise assignment of the beta-sheet secondary structure to the nano ribbons of the the A6K peptide, at this stage I recommend the paper for publication in the journal.

With my best regards.

Author Response

Dear reviewer,

Thank you for agreeing to our article for publication in the journal, and thank you for your valuable comments.

kind regards,

Sai Wen on behalf of the authors

Reviewer 3 Report

The authors have attended the entire minor revisions. However, three points still missing regarding major revisions. 

1 - As the peptide sample was acquired commercially, the authors need to inform the product purity.

2 - Item 3.1 – The authors must describe how the peptide concentration (16 mM) was determined. UV absorbance, Aminoacid analysis or simply Gravimetry?

3 -  Item 3.4 – Include the CD experiment set-up: scan speed, data pitch, response time and accumulations. 
